# Exploring disaster impacts on adaptation actions in 549 cities worldwide

Daniel Nohrstedt [1,2 ✉], Jacob Hileman [1,2], Maurizio Mazzoleni [2,3,4], Giuliano Di Baldassarre [2,4] & Charles F. Parker [1,2]

Whether disasters influence adaptation actions in cities is contested. Yet, the extant knowledge base primarily consists of single or small-N case studies, so there is no global overview of the evidence on disaster impacts and adaptation. Here, we use regression analysis to explore the effects of disaster frequency and severity on four adaptation action types in 549 cities. In countries with greater adaptive capacity, economic losses increase city-level actions targeting recently experienced disaster event types, as well as actions to strengthen general disaster preparedness. An increase in disaster frequency reduces actions targeting hazard types other than those that recently occurred, while human losses have few effects. Comparisons between cities across levels of adaptive capacity indicate a wealth effect. More affluent countries incur greater economic damages from disasters, but also have higher governance capacity, creating both incentives and opportunities for adaptation measures. While disaster frequency and severity had a limited impact on adaptation actions overall, results are sensitive to which disaster impacts, adaptation action types, and adaptive capacities are considered.

[1] Department of Government, Uppsala University, Uppsala, Sweden. [2] Centre of Natural Hazards and Disaster Science (CNDS), Uppsala University, Uppsala, Sweden. [3] Institute for Environmental Studies, Vrije Universiteit Amsterdam, Amsterdam, the Netherlands. [4] Department of Earth Sciences, Uppsala University, Uppsala, Sweden. ✉email: daniel.nohrstedt@statsvet.uu.se

Many cities around the world are projected to suffer increasing losses from extreme natural hazard-related disaster events, such as floods, storms, and wildfires[1]. The ability to adapt from past disasters will be crucial for enhancing local preparedness to future events and reducing disaster risk[2,3], making cities important centers of adaptation decision-making[4,5]. Although the impacts of disasters on local adaptation have been studied in various fields and cases[6–8], the relationship between disasters and policy action generally remains uncertain and contested[9–12]. Some studies suggest that disasters significantly shape adaptation[13–16], while others attribute adaptation to interactions between disasters and other factors[17–19], or find no relationship[20]. These findings, however, primarily are derived from single or small-N case studies, meta-analyses, and a few systematic studies of cities in Europe and North America[21–25] while evidence is lacking from other parts of the world.

Three theoretical perspectives of the disaster-adaptation action connection can be discerned from previous research (Fig. 1). The first perspective (Fig. 1a) depicts adaptation action as a response to the repeated occurrence of disaster events and their resulting human and financial costs. Frequent disasters and significant human and economic losses are expected to bring heightened public attention to problems and pressure for governmental action, increasing the probability that responsible policy-makers take adaptation action[17,26,27]. This is particularly likely at the local level, where policy-makers are relatively accessible to the public, making them potential targets for direct pressure from constituents[28]. Yet, major disasters often also spark blame and accountability battles, which can breed defensiveness and shift attention away from response performance. A typical reaction to

escape accountability is to deflect blame rather than drawing lessons[29], reducing the probability of adaptation action. Perceptions of past disaster responses as successes may also lower the urgency for adaptation actions[9]. It has also been suggested that higher impact events may overwhelm communities so that they are unable to address issues other than relief and reconstruction[30]. These assertions prompt exploration into the possible effects that the frequency and severity of disaster events might have on adaptation actions.

The second perspective (Fig. 1b) recognizes that disasters can provoke adaptation actions that differ according to underlying characterizations of risk[31]. Adaptation entails measures to reduce vulnerability and mitigate impacts of natural hazards[32,33], which span a wide range of actions to address different hazard types (e.g., both climatological and geological), general disaster preparedness, and other social and environmental issues not directly related to disasters. Adaptation actions can be sorted into four distinct types. Specific disaster adaptation includes actions taken to target future disaster events of the same type as those that have previously struck a locality, for example, adopting flood defense measures following a major flooding event. Expansive adaptation involves taking actions to prepare for future disasters not previously experienced, for instance, mapping urban heat islands even though heat waves have not historically been an issue. Generic preparedness refers to actions taken to strengthen readiness and response for any future disaster, regardless of type, such as crisis management planning, early warning systems, or establishing evacuation routes. Disasters may also incentivize actions associated with climate change and other sources of social and environmental harm unrelated to specific disaster types or

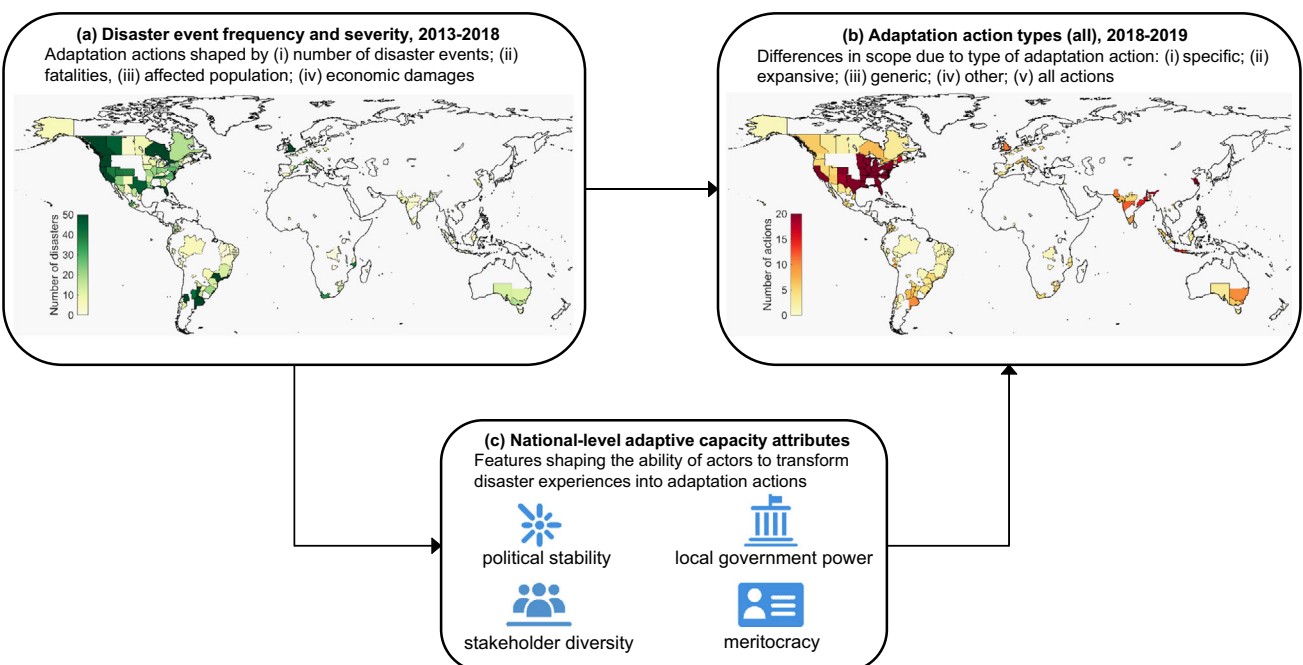

**Fig. 1 Theoretical perspectives linking disaster frequency and severity, adaptive capacity, and adaptation actions in cities.** Perspective (**a**) recognizes relationships between the number of disaster events, fatalities, affected population, and economic loss and adaptation actions. Perspective (**b**) focuses on potential correlations between disaster events and specific disaster adaptation actions, expansive adaptation actions, generic preparedness actions, and other actions. Perspective (**c**) acknowledges confounding effects of adaptive capacity attributes, including political stability, local government power, stakeholder diversity, and meritocracy. Panels a and b display the geographical locations of the disaster events and the total number of adaptation actions, respectively, included in the analysis. Maps in panels a and b created in Matlab R2021b with publicly available map data derived from Natural Earth, https://www.naturalearthdata.com/downloads/10m-cultural-vectors/10m-admin-1-states-provinces/, and https://sedac.ciesin.columbia.edu/data/set/grump-v1-national-admin-boundaries. Icons (panel **c**) source and credit: United Nations Office for the Coordination of Humanitarian Affairs, OCHA, https://brand.unocha.org/d/xEPytAUjC3sH/icons, under the Creative Commons License Attribution 4.0 International (CC BY 4.0), https://creativecommons.org/licenses/by/4.0/, material has not been modified.

**Table 1 Effects of disaster event frequency and severity factors (absolute measures) on types of adaptation actions in cities.**

|  | Specific | Expansive | General | Other | All |
|---|---|---|---|---|---|
| Event frequency | 0.615*** (0.143) | −0.231*** (0.069) | 0.067 (0.064) | 0.107 (0.077) | 0.557** (0.264) |
| Economic losses | 0.322** (0.134) | 0.092 (0.059) | 0.103* (0.053) | 0.072 (0.072) | 0.588** (0.239) |
| Affected population | −0.194 (0.219) | −0.070 (0.108) | −0.070 (0.097) | −0.235* (0.123) | −0.569 (0.412) |
| Fatalities | −0.086 (0.174) | 0.004 (0.082) | −0.031 (0.071) | −0.051 (0.093) | −0.164 (0.313) |

Coefficients for bivariate relationships between disaster frequency, economic losses, affected population, fatalities (all measured in absolute terms), and the number of adaptation actions by type. Robust standard errors in parentheses. *significant at $p < 0.1$; **significant at $p < 0.05$; ***significant at $p < 0.01$.

general disaster preparedness, such as biodiversity monitoring programs and economic diversification measures.

The third perspective (Fig. 1c) holds that the impact of disaster events on adaptation actions is conditioned by features of the political system that enable or constrain policy actors to constructively process and transform experiences from disasters into concrete adaptation actions. Which features matter remains contested and are context-specific, but most adaptive capacity frameworks[34–38] emphasize several attributes that vary across countries, including political stability, local government power, stakeholder diversity, and meritocracy. These attributes represent national-level opportunity structures mediating the effect of disasters on adaptation actions. These structures may also shape adaptation actions independently of disasters.

In this work, we explored these three perspectives by assessing potential relationships that disaster frequency, disaster severity, and adaptive capacity attributes have with different types of adaptation actions in 549 cities globally. We showed that the frequency and severity of major disasters generally have limited influence on adaptation actions in cities, although effects varied across action types, severity measures, and system attributes associated with adaptive capacity. Disaster event frequency, affected population, and fatalities tend not to affect adaptation actions. Economic damage, however, did predict some types of adaptation actions, but only did so under conditions of high levels of political stability and meritocracy. Societies with these attributes also recorded high monetary losses, indicating a wealth-effect; developed societies suffer greater economic damages from disasters but also have greater governance capacity to support adaptation decision-making. Our study also demonstrated that cities were likely to take adaptation actions if they were located in regions that experienced more recent disasters. In contrast, temporally distant events had no effect on adaptation actions.

## Results

This study combined data on 673 natural hazard events derived from the International Disasters Database (EM-DAT)[39] with data on 3,604 climate adaptation actions—representing 243 unique categories of adaptation action (Supplementary Table 2)—in 549 cities derived from the CDP (formerly Carbon Disclosure Project)[40] database. The EM-DAT disasters included in the study cover nine hazard types: drought, earthquake, extreme temperature, flood, landslide, mass movement, storm, volcanic activity, and wildfire. Since EM-DAT collects disaster event data at the country level, we followed previous work[41] and relied on geocoded information about the administrative units impacted by each hazard event to link event impacts to the sub-national regions where specific cities are located. In addition, the analysis explored relationships among adaptation actions reported by cities in 2018 and 2019, and the frequency and severity of disaster events that occurred in the five years preceding the actions (i.e., 2013–2017 for 2018 actions and 2014–2018 for 2019 actions, see Fig. 1a, Fig. 1b). Given the observation that the abnormality of weather events compared to historical patterns shapes attention

to climate change in general[42], we calculated disaster frequency and severity relative to 15-year country averages[11]. We also normalized frequency and severity measures against city population (for fatalities and affected population) and national-level indicators, including total disasters during the time-period of the study (for frequency) and Gross Domestic Product (GDP) per capita (for economic damages).

**Specific adaptation actions commonly follow major disasters.** We found that cities commonly reported taking disaster-specific adaptation actions following major disasters in the regions where each city is located. Of the 3,604 adaptation actions undertaken by cities in 2018 and 2019, 47% are specific disaster actions ($n = 1,691$), 17% are expansive disaster actions ($n = 624$), 16% are generic preparedness actions ($n = 575$), and the remaining 20% are other actions ($n = 714$). These adaptation actions were taken by cities from every major world region (Fig. 1b; Supplementary Figure 1), although over half are located in high-income countries.

**The influence of disaster frequency and severity vary by measure and adaptation action type.** We use linear regression to explore basic bivariate relationships between disaster event frequency and severity factors, and each adaptation action type. Table 1 shows that event frequency and economic damage, in absolute terms, significantly affect the number of adaptation actions. However, effects vary by adaptation action type and disaster attribute (full results for each model are provided in Supplementary Tables 4–8). Economic damage has a significant positive effect on adaptation actions regardless of type ('all actions', regression coefficient $b = 0.588$, $p < 0.05$), but especially specific adaptation actions ($b = 0.322$, $p < 0.05$). Disaster frequency has a significant positive effect on specific disaster adaptation actions ($b = 0.615$, $p < 0.01$) but a negative effect on expansive actions ($b = −0.231$, $p < 0.01$). Affected population has a weak negative effect on other adaptation actions ($b = −0.235$, $p < 0.1$), while fatalities do not have a significant effect on any type of adaptation action. However, fatalities do have significant effects in the models based on normalized measures of disaster frequency and severity (Supplementary Tables 9–13), but these effects are limited to weak negative correlations. The 15-year baseline measures (Supplementary Tables 14–18) have few significant effects, with the notable exception of disaster frequency on specific adaptation actions ($b = 0.845$, $p < 0.05$).

**Adaptation actions are unrelated to the combined effects of event frequency and severity.** We find little evidence that combinations of more frequent and more severe disasters shape adaptation actions. Combining disaster event frequency and severity measures in multiple regression models, including models with normalized and 15-year baseline measures (Supplementary Tables 24–26), only marginally improves model fit compared with the bivariate models. Model fit, however, remains low overall. The combined model (Supplementary Table 24)

examining the number of adaptation actions in cities suggests that the effect of disaster frequency on specific actions is weakened, possibly due to economic losses. Both these effects are significant ($p < 0.05$). Event frequency also remains negatively correlated with expansive adaptation actions, and this effect is slightly reinforced ($b = -0.307$, $p < 0.01$) in relation to the bivariate model, suggesting that an increase in the number of disasters that inflict significant economic losses reduces the number of adaptation actions aimed at disaster types other than those that recently occurred. However, we also tested the interaction term frequency × economic losses, which was not significant for any type of action (Supplementary Table 32).

### Effects of disaster frequency and severity on adaptation action depend on city size and political stability.

We next explored the potential mediating effects of national-level adaptive capacity attributes on adaptation actions in cities (box c, Fig. 1). Additionally, based on studies showing that climate action and support for climate adaptation action correlate with more recent events[43–46] and city size[47–49], we also added time lag (mean years between actions and disasters in a region) and city population size as control variables in the multiple regression models. Adding population size is also essential in order to account for the possibility that fatalities and affected populations are mechanically correlated with population[50]. We tested separate models with absolute, normalized, and 15-year baseline measures of disaster frequency and severity factors. The results (Fig. 2; Supplementary Tables 33–44) show that adding these controls weaken some effects of event frequency and severity factors on adaptation actions, while other effects disappear entirely.

First, the significant negative effect of event frequency on expansive adaptation actions remains but is most robust for the absolute number of disasters, and moderate for the 15-year baseline. An increase in the number of disasters thus reduces the number of adaptation actions targeting other event types than those previously affecting a region when controlling for adaptive capacity attributes, time lag, and city population size. Second, event frequency has no effect on specific adaptation actions in these models. Third, economic damages has a significant positive effect only in the case of specific actions, when measured as baseline damages, and in the case of expansive actions, when

measured as absolute damages. The results (Supplementary Tables 34, 38, 42) indicate that these changes are due to confounding effects of city size and political stability, which correlate positively with every adaptation action type, except actions in the category 'other'. These results suggest that cities with larger populations and cities that are located in more politically stable countries tend to undertake more adaptation actions regardless of the number and severity of disasters (Supplementary Table 28). We also find significant negative effects of stakeholder diversity in these models, but these are primarily associated with specific adaptation actions. Lastly, time-lag has a significant negative effect on specific and expansive actions in all models (Supplementary Tables 29–44), indicating that adaptation actions follow after more recent events.

### Adaptive capacity attributes moderate damage impacts.

We conducted moderation analyses to test whether effects of disaster frequency and severity on adaptation actions in cities systematically differed across levels of national adaptive capacity (Fig. 1c). Specifically, we conducted separate regressions focusing on significant bivariate relationships, which are limited to event frequency and economic damage—measured in absolute terms—as predictors of specific, expansive, and general adaptation actions. The pattern here is mixed: some adaptive capacity attributes moderate impacts of economic damages but not of frequency. Moreover, the influence of event frequency and damages on adaptation actions is largely unrelated to city population size.

To start with, results in Table 2 show that interaction terms combining disaster frequency with each of the four adaptive capacity variables are insignificant for all three adaptation action types, with the exception of weakly significant effects on expansive actions, suggesting that the effect of event frequency on adaptation actions overall is not mediated by any of the adaptive capacity attributes.

We get different results for interactions between economic damages and adaptive capacity. As shown in Table 2, while adaptive capacity attributes do not strongly mediate effects of economic damages on expansive adaptation actions, all interaction terms, except for local government power, are significant in relation to specific adaptation actions. Thus, the extent to which

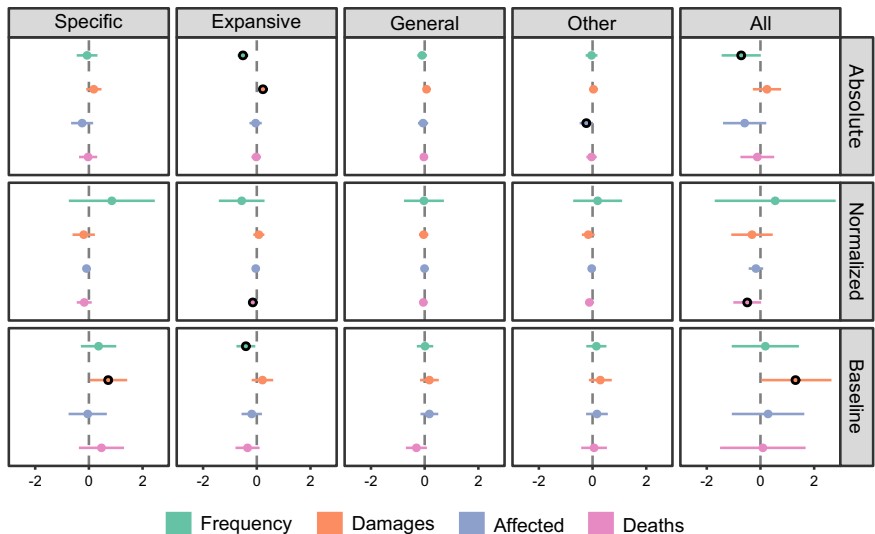

**Fig. 2 Estimated effects size.** Plots showing estimated effect size of absolute, normalized, and 15-year baseline measures of disaster event frequency, economic damages, affected population, and fatalities on adaptation action types. Estimates from multiple OLS regression models with time lag, city population, political stability, meritocracy, stakeholder diversity, and local government power as controls. Highlighted circles indicate estimates significant at $p < 0.1$ or better, while the error bars display the 95% confidence interval. Full models are shown in Supplementary Tables 33–44.

**Table 2 Results of moderation analyses, including interaction effects of event frequency, economic damage, city population, and adaptive capacity attributes on specific, expansive, and general adaptation actions.**

|  | (1) Specific | (2) Expansive | (3) General |
|---|---|---|---|
| Frequency × city population | 0.137 (0.092) | 0.074* (0.044) | −0.001 (0.041) |
| Frequency × political stability | −0.027 (0.115) | −0.012 (0.056) | −0.032 (0.052) |
| Frequency × meritocracy | 0.043 (0.461) | 0.179 (0.225) | −0.039 (0.210) |
| Frequency × stakeholder diversity | 0.582 (0.528) | −0.479* (0.254) | 0.104 (0.238) |
| Frequency × local government power | 0.164 (0.162) | −0.014 (0.078) | 0.018 (0.073) |
| Damage × city population | −0.006 (0.051) | 0.030 (0.022) | −0.007 (0.020) |
| Damage × political stability | 0.126** (0.057) | 0.001 (0.026) | 0.068*** (0.023) |
| Damage × meritocracy | 0.412*** (0.150) | −0.024 (0.068) | 0.125** (0.062) |
| Damage × stakeholder diversity | −0.472** (0.234) | 0.215** (0.102) | 0.299*** (0.093) |
| Damage × local government power | −0.049 (0.134) | 0.012 (0.058) | 0.002 (0.054) |

Coefficients for interaction terms of combinations of disaster event frequency, economic losses, city population, and four adaptive capacity attributes, including political stability, meritocracy, stakeholder diversity, and local government power. Frequency and losses are measured in absolute terms. All models include time lag (years since the most recent disaster event) and city population as control variables. Robust standard errors in parentheses. *significant at $p < 0.1$; **significant at $p < 0.05$; ***significant at $p < 0.01$. Complete models in Supplementary Tables 49, 50, 53, and 54.

cities take adaptation actions targeting disasters of the same type as those that recently occurred partially depends on whether these cities are located in politically stable countries and countries that value meritocracy. Political stability, meritocracy, and stakeholder diversity also mediate the effect of economic damages on general adaptation actions.

Furthermore, Table 2 shows that the interaction term economic damage × stakeholder diversity has a significant negative effect ($b = −0.472$, $p < 0.05$) on specific adaptation actions. This would suggest that cities in countries with higher stakeholder diversity in decision-making take fewer specific actions after disasters with greater economic costs. However, when analysed separately from economic damages, stakeholder diversity also had significant negative effects on specific adaptation actions (Supplementary Tables 27 and 28). It is thus possible that an increase in stakeholder diversity reduces the number of specific adaptation actions independently of economic damages inflicted by disasters.

We carried out simple slopes analysis to explore further the nature of these significant interactions. The results (Fig. 3) indicate that economic losses have significant positive effects on specific adaptation actions in cities located in countries at high levels of political stability ($b = 0.48$, $p < 0.05$, Fig. 3a) and meritocracy ($b = 0.63$, $p < 0.05$, Fig. 3b), but no significant effects in countries with lower levels of stability and meritocracy. We also find a weak positive effect from economic damages on specific adaptation actions at the mean level of meritocracy ($b = 0.29$, $p < 0.05$, Fig. 3b), but no significant effect at the mean level of political stability. In contrast, economic damage has a significant positive effect on specific adaptation actions at low levels of stakeholder diversity ($b = 0.70$, $p < 0.05$, Fig. 3c), but no significant effect at high levels of stakeholder diversity. Economic damages have significant effects on general adaptation actions at high levels of political stability ($b = 0.25$, $p < 0.05$, Fig. 3d), meritocracy ($b = 0.21$, $p < 0.05$, Fig. 3e), and stakeholder diversity ($b = 0.42$, $p < 0.05$, Fig. 3f), but no significant effects at low levels of these three variables.

## Discussion

Whether disasters incentivize policy actors in cities to take adaptation action is the subject of scholarly debate. This research represents a nascent, yet growing, study area[51] in need of empirical work to estimate the potential effects of disasters on different types of adaptation actions in cities worldwide. Our results suggest that major disasters generally play a limited role in

shaping adaptation actions in cities. Although we find several significant effects of disaster-related variables on adaptation actions, these effects are modest. Moreover, experience from past disasters only accounts for a small proportion of the variation in adaptation actions. Nevertheless, the study unveils previously unidentified impacts of disaster events on adaptation actions in cities.

One crucial contribution of this study is to disaggregate sub-categories of adaptation actions taken by cities to address disaster event types that recently occurred within the city region (specific adaptation); disaster types other than those recently experienced (expansive adaptation); efforts to strengthen disaster prepared-ness independently of disaster event types (general preparedness); and issues dealing with climate change and related social and environmental harms (other). Differentiating disaster event frequency and measures of severity, including human loss in terms of affected population and fatalities, and economic damage, enabled us to explore the individual and combined effects of disaster attributes on the adaptation action types, and to compare effects for different empirical measures. Overall, the results show that disaster event frequency and economic damages, measured in absolute terms, have some influence on the probability of certain types of adaptation action in cities. When only accounting for bivariate relationships (Supplementary Tables 4–18), event frequency had significant effects on specific adaptation actions, expansive actions, and general preparedness actions. However, the frequency and severity of disasters had no effect on other adaptation actions broadly targeting climate change and related social and environmental harms. Meanwhile, the study demon-strates that leaving out adaptive capacity attributes causes omitted-variable bias, which confirms that political institutions associated with adaptive capacity provide vital conditions that shape the ability of cities to transform disaster experience into policy action. This risk is evident in relation to disaster event frequency. In other words, if the effects of disaster event fre-quency on adaptation actions are assessed separately from adaptive capacity attributes, one could erroneously conclude that the number of disasters is a critical driver of different types of adaptation actions in cities.

Whether the frequency of disasters shapes adaptation actions is contested in previous literature. Our study demonstrates that an increase in the number of disasters is followed by a decrease in expansive adaptation actions (Table 1, Fig. 2). The frequency of disasters has no effect on specific adaptation actions, generic preparedness actions, or actions unrelated to disasters (Fig. 2). These results hold across absolute and 15-year baseline measures

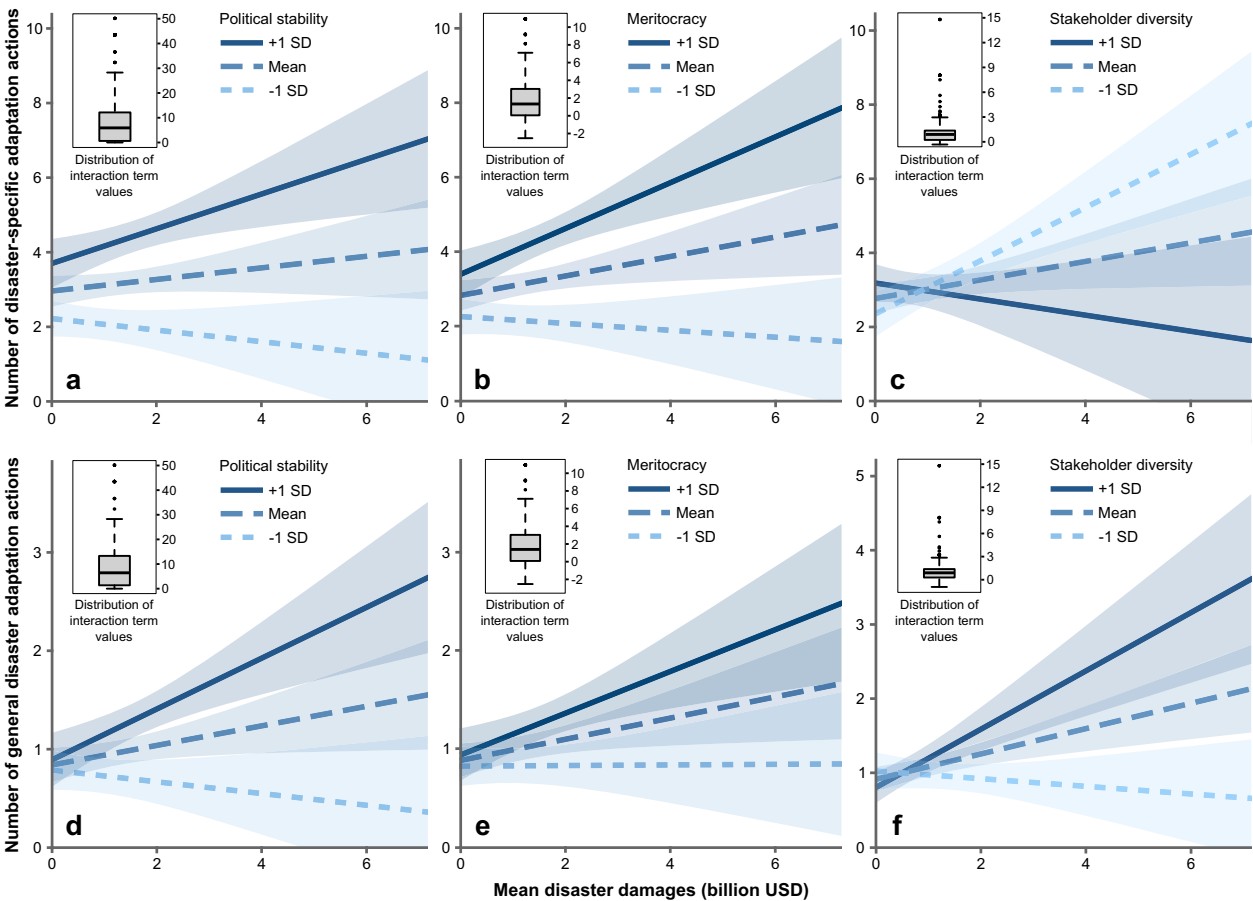

**Fig. 3 Results of simple slopes analyses.** Top row panels show results of simple slopes analyses of the strength of the relationship between economic damages and specific adaptation actions at high (+1 SD), mean, and low (−1 SD) levels of political stability (**a**), meritocracy (**b**), and stakeholder diversity (**c**). Bottom row panels show the relationship between economic damages and general adaptation actions at high (+1 SD), mean, and low (−1 SD) levels of political stability (panel **d**), meritocracy (**e**), and stakeholder diversity (**f**). The coloured bands denote the 90% confidence interval.

of disaster frequency, and when controlling for the number of years since the last disaster, city population size, and adaptive capacity factors (Supplementary Tables 29–33). Thus, the evidence suggests that, on average, an increase in the number of disasters affecting cities reduces the number of actions to strengthen preparedness to other hazard types than those that were experienced directly. We see this result as intuitive since we would expect that disaster events are less relevant as a trigger for adaptation in policy areas that are not immediately affected[52]. Based on previous studies[53,54], we also speculate that generic preparedness, unmoored to threats from a specific hazard type, is indistinct and, therefore, rarely becomes subject to stakeholder mobilization or public pressure for change in the wake of disasters. That event frequency did not have an effect on 'other' actions, which here refer to adaptation measures largely unrelated to disasters, is also not unexpected in light of prior conflicting evidence concerning impacts of disasters on general attitudes to climate change[55,56]. These results confirm the importance of clearly distinguishing subcategories of adaptation actions.

Although detailing potential causal logics that underpin these effects is beyond the scope of the study, models with national-level adaptive capacity attributes (Supplementary Tables 19–23, 28) suggest that cities with bigger populations and cities that are located in countries with greater levels of political stability, meritocracy, and stakeholder diversity, on average, report more adaptation actions of all types except actions unrelated to disasters (category 'other'). It is theoretically possible that these features help ensure that actors with the ability to acquire and

utilize knowledge gain influence in adaptation policy-making. Moreover, individuals and organizations with these skills are generally responsive and capable of building consensus and shaping policy agendas, which may be vital qualities that enhance adaptation action in the aftermath of disasters[57]. In addition, adaptive capacity attributes provide important opportunity structures that enable or constrain advocacy coalitions to influence agendas and decision-making in the aftermath of disaster[58]. These factors deserve more empirical research[27], including the in-depth examination of post-disaster policy processes in selected cities.

Our investigation of potential interaction effects suggests buffering from some of the country-level attributes, but these effects only apply to economic damages. We found only a few weak interaction effects involving disaster frequency. The pattern (Table 2, Fig. 3) is that economic damages from disasters – when measured in absolute monetary terms – influence some adaptation actions (disaster-specific and general preparedness actions) in cities located in countries with higher levels of political stability and meritocracy. No significant effects of economic damage on adaptation actions were found among cities located in countries with lower levels of political stability and meritocracy. Buffering effects of stakeholder diversity, measured by the width of stakeholder consultation in relation to policy change at the national-level, are inconsistent; increased economic damage led to more specific adaptation actions in countries with lower levels of diversity, but more general actions in countries with higher levels of diversity.

We hypothesize that these patterns might be the product of a wealth effect. In these data, cities that are located in countries with more adaptive capacity attributes (+1 SD, Fig. 3) on average suffered greater absolute monetary losses from disasters than cities in countries with fewer adaptive capacity attributes. The average economic damages affecting cities in countries with high political stability, for instance, were significantly larger than in countries with medium and low levels of political stability. It is important to note here that economic damages in the EM-DAT dataset represent the total disaster damages recorded in a country, not in an individual city, although the geocoded data likely connect some proportion of damages to cities. Out of 229 cities located in countries with higher political stability, 227 were located in high-income countries and 2 in upper-middle countries, according to the World Bank classification. Our data, thus, suggest that income-level is an important confounder that drives both losses and adaptive capacity; developed countries suffer greater absolute monetary losses but also have more governance resources to support adaptation decision-making at different levels.

Results concerning stakeholder diversity provide another illustration of a potential wealth-effect, but here we found two different effects for different adaptation action types. These effects were noted exclusively for cities in high-income countries. The group of cities with high stakeholder diversity levels is roughly the same as for political stability; all 65 cities with high stakeholder diversity are located in high-income countries. Average losses for these cities were 0.5 billion USD. In this group, disaster losses were associated with an increase in the number of generic preparedness actions (Fig. 3e) but a decrease in specific actions (Fig. 3c). There is theoretical support for both these scenarios; diversity ensures the inclusion of more knowledge and beliefs in policy-making but also increases the likelihood that veto-players and conflicts delay or block adaptation actions.

We encourage future work to examine the wealth-effect in greater detail, particularly efforts to explore other data sources to document local disaster impacts and local variations in adaptive capacity. Such data sources are crucial to obtain a more detailed overview of how adaptive capacities may shape post-disaster policy-making at different levels of development and in different parts of the world. Here one intriguing avenue involves comparisons between cities with low levels of adaptive capacity, particularly in the Global South, where disasters have led to different policy reactions.

Our study demonstrates that fatalities and the number of people affected by disasters, regardless of how these impacts are measured, do not generally influence any of the adaptation action types. This finding, thus, suggests that the human toll of disasters is not a significant determinant of adaptation actions in cities across the globe. In addition, there is little evidence of any interaction effects of disaster frequency and severity measured both as economic and human losses. Contrary to some previous research that reports positive effects of repeated high-impact events on local adaptation action[53,59,60], our data show that after controlling for time lag, population size, and adaptive capacity, adaptation actions are largely unrelated to an increase in the number of high-impact events (Supplementary Table 32). Consequently, this study suggests that while disasters may become more frequent and severe, this combination per se appears to be insufficient as a catalyst for adaptation action in cities.

These results feed into the ongoing scholarly debate concerning the role and influence of disaster severity in shaping adaptation action. In this regard, we make two contributions. First, whereas some previous studies have examined fatalities as part of additive indices with other indicators (e.g., presidential disaster declarations)[30] or focused on perceived severity[61], we demonstrate the value of exploring effects of fatalities and other severity measures separately. Second, exploring different severity measures across adaptation action types can unveil more specific relationships, leading to new insights about potential causal mechanisms. Although prior literature[26,51,52,62,63] has come a long way in proposing potentially crucial drivers of policy action after disasters, focusing on, e.g., agenda-setting, stakeholder mobilization, and learning, these drivers could be specified further. For example, disasters that inflict significant economic damage might provoke mobilization of powerful economic interests, whereas victims' groups might be less resourceful and influential in initiating policy action. One may also find that conditions for post-disaster learning and change vary considerably across types of adaptation. Actions to strengthen generic disaster preparedness, for instance, may generate low public interest compared to adaptation actions targeting specific hazards. Specifying and empirically testing these and other mechanisms is an essential next step in this work.

Although the study spans four adaptation action types in cities from every major world region, there are aspects of this relationship that our analyses do not cover. There are also data-limitations that constrain the analysis in meaningful ways. While the EM-DAT dataset remains one of the richest and most widely used datasets of major disaster events[64], it excludes less consequential events with the potential to spark adaptation. Also, the study design assumes that disasters affecting cities in one period (i.e., 2013–2017 or 2014–2018) shape adaptation actions in the subsequent period (i.e., 2018 or 2019), which excludes short-term adaptation triggered by disasters occurring within the same year. This also brings a risk for spurious relationships in cases where several years unfolded between the most recent disaster event and the year for reporting adaptation actions. A related limitation is that the CDP data does not specify the date when each action was initially adopted. Nevertheless, we find significant negative effects of time-lag (Supplementary Tables 29–44), which suggest that adaptation actions are more likely after recent events. This finding corroborates results reported elsewhere concerning disaster impacts on climate change perceptions[43–46] and indicates that disasters open temporary opportunities for adaptation actions that fade with time.

It should also be noted that the study results depend on accurate raw data in the EM-DAT dataset about the geographic location of disaster impacts, which we relied on for geocoding. Although we took several steps to remove uncertain disaster event locations (see Methods), spatial resolution and coverage issues are generally acknowledged but not yet sufficiently addressed by existing disaster event datasets[41,65]. Lastly, we included national-level adaptive capacity features and controls. Although these are country-level measures with slight expected local variation, ideally sub-national indicators, which are currently unavailable for the sample used in this study, should be employed to eliminate risks for measurement error[59]. Also, other social, political, and institutional characteristics may make city-level policy actors more prone to exploit disaster events for adaptation action. Studies suggest, for instance, that socio-economic characteristics of communities, which shape the capacity of local leaders to take adaptation action[66], and governance capacities supporting regular and impartial monitoring, evaluation, and policy learning[67], are potentially important confounders to consider in future research.

It has been reported that international policy regimes for reducing disaster losses, such as the United Nations Hyogo Framework, have had limited local impact, making city actions essential for strengthening community preparedness to disaster[68]. These actions can be strengthened by exchanging experiences via transnational local-level climate networks (e.g., C40, Climate Alliance, Resilient Cities campaign, Covenant of Mayors), regarding the determinants of post-disaster policy-making.

Disasters are generally assumed to enable reforms that are not otherwise possible, which is the founding premise of the 'build back better' approach to more resilient communities. The results of this study corroborate analyses showing that disruptive disasters are not regularly leveraged as opportunities for adaptation[69,70]. However, this study suggests that this is not solely due to lack of capacity but is also a function of more complex relationships between different adaptation action types and disaster impacts.

## Methods

**Data sources and sample selection.** Data for this study were primarily derived from two freely available and established sources: (i) the International Disaster Database[39] (EM-DAT), and (ii) the CDP cities adaptation actions dataset[40] (formerly Carbon Disclosure Project, CDP). We derive disaster frequency and severity measures from the EM-DAT dataset and use the CDP dataset to identify different types of adaptation actions.

The EM-DAT dataset is a continuously updated catalogue of natural hazard events around the world. The dataset includes the type of hazard, spatio-temporal information, and the severity of the human and economic impacts, among other information. In order for a natural hazard event to be included in the EM-DAT dataset, it must have resulted in at least one of the following: (1) ten or more fatalities; (2) one hundred or more affected individuals; (3) a declaration of a state of emergency; or (4) a call for international assistance. As such, the EM-DAT dataset is not a complete catalogue of all natural hazard events, but rather provides a robust recording of large-scale events[71].

The CDP maintains an extensive repository of climate-related datasets, including adaptation actions taken by cities around the world. We specifically use the 2018–2019 cities adaptation actions dataset, which includes the city reporting the action(s), the year the action(s) occurred, and descriptions of the action(s) taken, among other information. This dataset contains actions that cities have disclosed to the CDP, or one or more global fora (e.g., Local Governments for Sustainability, Global Covenant of Mayors, C40 Cities) that provide data to the CDP. While the CDP describes the implementation status for each action, it does not provide a date when each action was initially adopted. Thus, the data explored here span actions adopted over 2018–2019, which may include the continuation of actions adopted further back in time.

The sample of natural hazard events included in the analysis is defined by the timeframe of the 2018–2019 CDP adaptation actions dataset. While the EM-DAT dataset includes events going back to the early 1900s, we focus only on the 2013–2018 period, as this provides the basis for assessing potential short-term lagged effects between disasters and adaptation actions. However, we draw on data from 1998 to 2012 in the EM-DAT dataset to establish baselines for evaluating the relative frequency and severity of the natural hazard events included in our sample. We also include a measure of the number of years between events and adaptation actions for each city to control for potential lagged effects. Furthermore, although this is a global study, we only examine countries ($n = 69$) that are both included in the CDP dataset (i.e., countries for which we have data on city-level adaptation actions over 2018–2019) and the EM-DAT dataset (i.e., countries that experienced one or more large-scale disasters during 2013–2018).

We also include a series of country-level control variables (see Supplementary Table 3 for full descriptions of all variables) derived from various sources of data:

- To normalize disaster frequency, we use the ratio of the disaster events that impacted each subnational administrative unit to the total events experienced at the country level during the same time period.
- To normalize disaster severity—economic damages, affected population, and fatalities—we use 2017 GDP per capita (for damages) and city population from the CDP dataset (for affected population and fatalities).
- To control for variation in overall adaptive capacity across countries, we include measures of political stability, diversity of stakeholders engaged in policy-making, meritocracy, and local government power from the Varieties of Democracy and Quality of Government datasets.

While we control for various country-level attributes that can support or hinder adaptation actions, we note the cities that reported actions during 2018–2019 are predominantly located in high income ($n = 274$) and upper-middle-income ($n = 208$) countries. Considerably fewer cities in the dataset belong to lower-middle-income ($n = 57$) and low income ($n = 10$) countries. In spite of the geographic variation of the cities across all major world regions, there are economic equity issues embodied in the CDP dataset. However, the relationship between disaster frequency and severity and adaptation actions remains ambiguous even when accounting for these differences.

**Coding protocol and procedures.** The dependent variable in this study is the number of adaptation actions reported to the CDP by each city over 2018–2019, which we classified into four distinct types. The predictor variables are characteristics of the natural hazard events documented in the EM-DAT dataset from 2013–2018 that affected the regions (i.e., first-order administrative units) where

these cities are located. Predictor variables include hazard frequency (i.e., the number of events over the analysed period) and severity (i.e., the average economic damages, affected population, and fatalities). In order to perform the analysis, we first needed to integrate the EM-DAT and CDP datasets, which required developing a coding protocol for linking natural hazard events with adaptation actions. This protocol allows us to: (a) determine whether a particular adaptation action addresses one or more types of natural hazard event; (b) determine whether an individual adaptation action and observed natural hazard event occurred in the same geographic region; and (c) determine the type of adaptation action based on (a) and (b) above. Significantly, this protocol minimizes the potential for spurious correlation between disaster events and adaptation actions in several key ways[7].

a)  *Classifying adaptation actions by natural hazard types.* This step of the data coding involved reviewing each unique adaptation action description in the CDP cities adaptation actions dataset, and determining whether the action was related, via direct effects, to one or more of the nine hazard types recorded in the EM-DAT dataset: drought, earthquake, extreme temperature, flood, landslide, mass movement, storm, volcanic activity, and wildfire. For example, while wildfires often coincide with droughts and heatwaves, we only consider wildfire-related adaptation actions directly related to wildfire events. The result is a series of 1 s and 0 s indicating which types of disasters a given action does and does not address (Supplementary Table 2 provides the full output from this step of the coding protocol). In addition, we conducted an inter-coder reliability test of 20 randomly selected adaptation action descriptions, which indicated over 70% agreement among three coders (Supplementary Figure 2).

b)  *Geocoding adaptation actions and disaster events.* This data coding step involved reviewing the geographic information provided for both the adaptation actions and disaster events, and determining the region (i.e., first-order administrative unit) where they occurred. Given that the CDP adaptation actions dataset provides both the country and city names, the geocoding process was a straightforward matching using a global list of first-order administrative units, which was compiled from two datasets: "Admin 1 – States, Provinces" from the Natural Earth map data and ArcGIS "World Administrative Divisions" map package. The EM-DAT dataset provides country names, along with a non-uniform, subjective description of the subnational areas impacted by a given disaster event. In preparing the data for this study, we manually reviewed the location descriptions in EM-DAT to determine the first-order administrative unit(s). This step was necessary to ensure a reasonable level of proximity between the locations of disaster events and cities reporting adaptation actions.

c)  *Identifying adaptation action types.* We define four general types of adaptation actions, based on the type(s) of natural hazard a specific adaptation action addresses, and whether or not one or more of these natural hazard types occurred in the region where a given action was reported. Specific disaster adaptation actions address one or more hazard event types that occurred in the region where a city is located. Expansive adaptation actions address one or more hazard event types that did not occur in the region where a city is located. Generic preparedness actions constitute actions that are not specific to any particular hazard type, but represent more general disaster risk reduction strategies (e.g., public awareness campaigns, crisis management planning). We also considered other actions to be those that did not fit the description of any of the three types. In addition, we ran models based on all actions, i.e., regardless of type, to assess how the results might differ if adaptation actions are not disaggregated into clearly defined subcategories.

**Assessing the influence of disaster event frequency and severity on adaptation actions.** We first perform a series of bivariate regression analyses (Supplementary Tables 4–8) to assess the extent to which more frequent and severe disaster events (independent variables) are associated with higher numbers of reported adaptation actions within each type, respectively (dependent variable). We additionally consider baseline measures of frequency and severity at the country level, which we calculate as the three-year running average of these metrics for all events in the EM-DAT dataset, within a given country, over a 15-year baseline from 1998 to 2012 (i.e., the average of 1998–2000, 1999–2001, 2000–2002, etc.). We adjust all economic damages for inflation, relative to the year 2021, using the Consumer Price Index dataset in order to ensure damages are comparable over time. Establishing baselines for hazard frequency and severity in this manner allows us to determine whether the hazards from 2013–2018 in our dataset deviate considerably from the recent historical average. We also utilize several control variables to account for variations in overall adaptive capacity and national development (Supplementary Table 3).

Next, we define multiple regression models, including all relevant independent and control variables. We include four measures of national-level attributes shaping the ability of a system to undertake adaptation actions: existence and power of local government, political stability, meritocracy, and that there was a variety of actors engaged in policy-making. The average number of years elapsing between disasters and adaptation actions is included to test whether a time lag affects the likelihood that cities will act. Lastly, we perform simple slopes analysis to

investigate significant interaction effects (using Johnson–Neyman intervals) between economic damages, adaptive capacity measures, and select adaptation action types.

We examined variance inflation factors (VIFs) for all multiple regression models in order to assess whether multicollinearity was an issue. No variables had VIFs greater than five in any of the models without interaction effects, which indicates that multicollinearity is within acceptable levels.

## Data availability

All data used in this research is publicly available via online sources. Data concerning disaster events are available from the International Disaster Database (EM-DAT) (https://www.emdat.be). Data for adaptation actions in cities can be accessed via the CDP (formerly Carbon Disclosure Project) cities adaptation actions dataset, (https://data.cdp.net/Adaptation-Actions/2018-2019-Cities-Adaptation-Actions/6zdf-pmyg). Data on Gross Domestic Product (GDP) per capita and country population is available at the World Bank (https://data.worldbank.org/), and data for adjusting US dollar values for inflation is available at (https://www.bls.gov/cpi/tables/supplemental-files/historical-cpi-u-202202.pdf). Data for political stability is available through the Quality of Government dataset (standard version) (https://www.gu.se/en/quality-government/qog-data/data-downloads/standard-dataset). For data about meritocracy, stakeholder diversity, and local government power, refer to the Varieties of Democracy dataset (version 11.1) (https://www.v-dem.net/en/data/data/v-dem-dataset-v111/). Data on first-order administrative units, which were used to plot provinces reported in the CDP dataset, rely on two publicly available datasets: "Admin 1 – States, Provinces" from Natural Earth map data (https://www.naturalearthdata.com/downloads/10m-cultural-vectors/10m-admin-1-states-provinces/) and "National Administrative Boundaries, v1 (2000)" from the Global Rural-Urban Mapping Project (GRUMP), v1 (https://sedac.ciesin.columbia.edu/data/set/grump-v1-national-admin-boundaries). The data generated from these sources for this study are provided in the Source Data file deposited in the following public repository: https://www.statsvet.uu.se/research/trampoline/data-repository/ Source data are provided with this paper.

## Code availability

The custom code and mathematical algorithm generated for this study have been deposited in the following public repository: https://www.statsvet.uu.se/research/trampoline/data-repository/

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

## Acknowledgements

The study was supported by the Swedish Research Council through a grant (No. 2018-03977) to a project entitled 'The transformative potential of extreme weather events' (TRAMPOLINE). All authors recognize the support from the Centre of Natural Hazards and Disaster Science (CNDS). G.D.B. and M.M. received additional support from the European Research Council (ERC, H2020 Excellent Science, Consolidator Grant No. 771678) to G.D.B.'s project entitled 'HydroSocialExtremes: Uncovering the Mutual Shaping of Hydrological Extremes and Society'.

## Author contributions

All authors designed the study; D.N., J.H., and C.F.P. developed the conceptualization; J.H. and M.M. collected the data and conducted data analyses; figures were created by D.N., J.H., and M.M.; D.N., J.H., C.F.P., M.M., and G.D.B. all contributed to the interpretation of the study results and writing the manuscript.

## Funding

## Competing interests

The authors declare no competing interests.
