## [Peer review file · Nature Communications]

Peer review comments first round -

Reviewer #1 (Remarks to the Author):

It gives me the pleasure to congratulate the authors on the work done which is commendable, especially for exhibiting originality and novelty. For instance, authors adopted the inter-coder reliability test, baseline assessment, trio-theoretical, methodological hybrid adoption, multi-dataset integration and exploration of the four adaptation forms is great but depicts the complexity embedded in unpacking adaptation actions adopted by selected 640 cities globally. I there suggest the following to improve the quality of the manuscript:

- Title could be modified to capture the number of cities upon which the analysis was based.

- Abstract: split the very long sentence (lines 13 -17) for clarity; highlight methodological aspects and the trio-theoretical perspectives adopted; Provide the number of these politically stable countries to qualify the statement (Lines 18 -19); Provide a concluding statement the offers the general implications of your findings at scales relevant to the study. Finally, ensure the keywords are listed.

-introduction (subheading is missing); Fig 1 - ensure that the words are eligible. Therefore, improve on the fonts to ensure eligibility. The study is generally situated in the existing body of knowledge of climate change-related disasters and adaptation in the global city space.

-Methods: The methods section should follow immediately after the introduction and thereafter, the results, discussion, and conclusion. This needs to be addressed in line with the journal format. This n= 58 countries contradicts the highlighted number 87 countries in the abstract and introduction section. Lines 340 -341: Is the countries' classification as the UN/World Bank or study? There is a need for clarity to provide the context. Lines 363 - 364: This is not very clear given the number of cities considered using the CDP dataset as being 492. Therefore, are the coded unique adaptation action descriptions equal to 276 for the CDP cities (492)? Lines 406 -407: provide the rationale for running additional models based on all actions, i.e., regardless of type? Line 432: what VIFs abbreviation stand for? This has not been provided in the sections above. To this end, I find the methodology robust, meeting the desired standards, and well-aligned to the nature of the study, which offers an opportunity for the replication of the approach at a microscale.

-Results, Discussion, and Conclusion:

The result section is well presented based on robust analysis and presented in a scholarly sound manner. Therefore, the work clearly supports the arguments advanced and the discussion. However, there is a need to have a separate conclusion section to clearly advance the conclusions based on the findings, context of the study and knowledge advancement, with clear spelt out recommendations and future research agendas. Having this section will also offer the readers to have an understanding of the synergies that exist between the adaptation forms i.e., specific, expansive, generic and others. There need also to highlight the limitations i.e., with the datasets used etc. to make future researchers be aware of.

In general, the results of the manuscripts disaster frequency and loss are fundamental in shaping adaptation measures depending on the context of the city as determined by the existing complex relationships prevailing. Therefore, there is no universal common approach that is adopted by cities as an adaptative strategy but a piecemeal combination. The study results are fundamental in

advancing research and debates in the climate change disaster and risk adaptation space especially in the global south which is not properly represented in the analysis.

NB; The highlighted aspects need to be addressed by authors to ensure that journal standards are maintained and met as required.

Reviewer #2 (Remarks to the Author):

Recommendation: Major revision

Thank you for the opportunity to review “Disaster frequency and loss shape hazard-related adaptations in cities.” This is a very interesting paper that addresses a major challenge and persistent gap for the adaptation literature – comparative explanatory research on adaptation responses, particularly at the local level. Nonetheless, there are two major issues here related to the time horizons in the dataset and the statistical models that have significant implications for the findings and require further attention. The following recommendations concern the formulation of the IV and DV data, and the types of models used in the analysis.

Time horizons:

Why these particular years for the disaster data? The time horizon is very short – 1 year following the disaster period is likely too short in most cases to see the adoption and implementation of substantive adaptations, which typically take longer to achieve (e.g. such as actions 7-9 & 13, 14, 16, 18 in Table A1). Planning cycles alone generally run longer than 1 year for those types of initiatives-implementation can indeed take many years in the case of major adaptation projects like infrastructure construction or upgrades. This implies that many (perhaps most) adaptations occurring in 2018-2019 are not occurring in response to events that happened in 2015-2017, and/or that they were initiated prior to 2018. Furthermore, this creates a potential bias in the dependent variable, particularly for expansive adaptations that may take longer to plan and implement given the challenges of post-disaster recovery (which would be expected to focus on top-of-mind hazards rather than other potential hazards). Related to this, the application of the 15-year baseline here only measures more severe/frequent disaster events in 2015-2017. If there were more significant events in the proceeding 5 years (that may have motivated adaptations implemented in 2018-2019), then it will seem as though cities actually have a lower frequency and severity of event experience in the current years of interests. The authors point to this as a limitation on page 12, however this is a significant issue (it speaks directly to the validity of the results) and it needs to be formally addressed in the analysis. What happens to the results if a longer time window of disaster data is used? E.g. 2010-2017 vs. a baseline of 2000-2010 (or even 2000-2017 and a baseline of 1990-1999).

Why these particular years for adaptation data? Are the data from CDP only adaptations that were taken up in that period, or are they all the adaptations that were in place during that period but may have been adopted earlier (or that were just registered in the CDP database during those years)? If it's the latter, then those earlier adaptations should be removed if the DV focus remains on actions adopted in the 2018-2019 period.

Overall the study needs more refinement around these issues, and sensitivity testing of the results to changes in the disaster events time horizon.

Models:

There's a question of scale in the statistical models used here- the DV data is local, the disaster data is regional and national, and the adaptive capacity data is national. Why didn't the authors use multi-level modelling here to account for the hierarchical nature of the data (particularly the country-level indicators)? I advise them to run a separate set of MLMs and report on whether these findings differ from the current models.

Second, why not control for local population, as it's shown in the literature that large cities tend to do more adaptation (quantity of adaptation actions) and more substantive adaptation (e.g. Araos et al. 2017, Lesnikowski et al. 2020)? This pattern is found across high, middle, and low-income countries. Only controlling for GNI misses this variation within countries.

Other comments:

Were adaptation actions related to non-climate change hazards removed from the dataset? On page 2 of the supplemental materials it's implied that hazards/responses related to earthquakes and volcanic activity are left in. These should be removed from the dataset if this is the case (over-reporting is a common challenge with country-driven reporting on adaptation, so responses to those types of hazards may make it into the CDP database but aren't aligned with conventional definitions of key climate risks, e.g. from the IPCC). If they were removed prior to analysis, then this should be specified on page 2.

The paper observes that "increased economic losses from disasters generally are followed by an increase in the number of specific disaster adaptation actions. Increased economic losses are also associated with an increase in other climate adaptation actions unrelated to disaster preparedness (subcategory 'other'). Economic losses do not influence the other adaptation action subcategories. These are robust effects that hold despite variation in adaptive capacity (Supplementary Table 21a)" (page 11). This is very interesting given previous research (e.g. Hallegatte et al. 2013) that finds economic damages are higher in developed countries and human damages are higher in developing countries, which is attributed to generally higher investment and resilience in infrastructure and social systems in wealthier places. Because the adaptive capacity data is national in this study, it's difficult to draw conclusions from this difference in findings. Controlling for local population as suggested above would help to facilitate comparison to these other studies. Do economic losses drive specific disaster adaptation actions everywhere, or is this relationship being obscured here by the tendency for higher economic losses to occur in large cities with more costly infrastructure and economic activity?

Supplemental materials addition: Please include a list or map of the cities in the dataset, and information on country/region membership.

Response memo, manuscript NCOMMS-21-24977-T

Dear Reviewers,

Thank you for the opportunity to consider our revised manuscript NCOMMS-21-24977-T, with the revised title ‘Disaster frequency, severity, and adaptation action in 549 cities around the world’. Below we provide a point-by-point response to your comments. As requested in the decision letter, the manuscript has undergone a major revision. In the revised version of the manuscript, we have highlighted in yellow the specific passages in the paper where we have addressed the reviewers’ concerns.

Reviewer 1 (R1)

R1.1. It gives me the pleasure to congratulate the authors on the work done which is commendable, especially for exhibiting originality and novelty. For instance, authors adopted the inter-coder reliability test, baseline assessment, trio-theoretical, methodological hybrid adoption, multi-dataset integration and exploration of the four adaptation forms is great but depicts the complexity embedded in unpacking adaptation actions adopted by selected 640 cities globally. I there suggest the following to improve the quality of the manuscript:

Authors’ response: Thank you for the constructive comments, which we address below.

R1.2. Title could be modified to capture the number of cities upon which the analysis was based.

Authors’ response: We followed this suggestion by revising the title to include the number of cities in the study.

R.1.2. Abstract: split the very long sentence (lines 13 -17) for clarity; highlight methodological aspects and the trio-theoretical perspectives adopted; Provide the number of these politically stable countries to qualify the statement (Lines 18 -19); Provide a concluding statement the offers the general implications of your findings at scales relevant to the study. Finally, ensure the keywords are listed.

Authors’ response: The abstract has been re-written for clarity and states that the paper explores the effects of disaster frequency and severity on four adaptation action types in 549 cities. The findings and general implications are also summarized. The revised version of the manuscript addresses the remaining points. For example, the number of countries at each level of adaptive capacity is now detailed in the revised Discussion section (p. 10). *Nature Communications* articles do not have keywords, so we have not included them.

R1.3. -introduction (subheading is missing); Fig 1 - ensure that the words are eligible.

Therefore, improve on the fonts to ensure eligibility. The study is generally situated in the existing body of knowledge of climate change-related disasters and adaptation in the global city space.

Authors' response: The heading to the introductory section has been added. Fig 1, including the legend, has been updated and fonts adjusted to ensure legibility.

R1.4a. Methods: The methods section should follow immediately after the introduction and thereafter, the results, discussion, and conclusion. This needs to be addressed in line with the journal format.

Authors' response: The revised version has been structured strictly according to the journal's formatting instructions (introduction; results; discussion; methods).

R1.4b. This n= 58 countries contradicts the highlighted number 87 countries in the abstract and introduction section.

Authors' response: Yes, there was a discrepancy in the first draft, which occurred because we were unclear in the first version about the difference between the total number of countries that were in the dataset compared to the actual number of countries we included in our sample. In the revised version (p. 12), we clarified that the number of countries covered in our sample is 69.

R1.4c. Lines 340 -341: Is the countries' classification as the UN/World Bank or study? There is a need for clarity to provide the context.

Authors' response: Correct, this is from the World Bank classification, which is now clarified in the revised version (p. 10).

R1.4d. Lines 363 - 364: This is not very clear given the number of cities considered using the CDP dataset as being 492. Therefore, are the coded unique adaptation action descriptions equal to 276 for the CDP cities (492)?

Authors' response: Thank you for flagging this confusing passage. We have edited the text (p. 3) to clarify that the study includes a subsample of the CDP data – 3,604 actions, reported by 549 cities, comprising 243 unique categories of adaptation action. We also refer readers to Supplementary Table B1, which includes the list of the adaptation actions. We further clarify (p. 12) that the reason we focus on a subset of the full CDP dataset is due to the fact that we only include those cities in the CDP data that correspond to the geographic scope of disasters recorded in the EM-DAT dataset during the 2013-2018 time-period of the study.

R1.4e. Lines 406 -407: provide the rationale for running additional models based on all actions, i.e., regardless of type?

Authors' response: In the revised version, we have added the rationale for models based on all actions in the methods section (see Methods, p. 14: Coding protocols and procedures, point c).

R1.4f. Line 432: what VIFs abbreviation stand for? This has not been provided in the sections above. To this end, I find the methodology robust, meeting the desired standards, and well-aligned to the nature of the study, which offers an opportunity for the replication of the approach at a microscale.

Authors' response: VIF stands for variance inflation factor, which is now written out in full the first time it is used (p. 15). We also have added a more thorough description of the VIF scores in the methods section (p. 15). We are pleased you found the methodology robust.

R1.5. Results, Discussion, and Conclusion: The result section is well presented based on robust analysis and presented in a scholarly sound manner. Therefore, the work clearly supports the arguments advanced and the discussion. However, there is a need to have a separate conclusion section to clearly advance the conclusions based on the findings, context of the study and knowledge advancement, with clear spelt out recommendations and future research agendas. Having this section will also offer the readers to have an understanding of the synergies that exist between the adaptation forms i.e., specific, expansive, generic and others. There need also to highlight the limitations i.e., with the datasets used etc. to make future researchers be aware of.

Authors' response: Thank you for these valuable suggestions on how to highlight the conclusions and implications of the study more clearly. We have attempted to do so in the revised Discussion section p. 9-11. In the revised version of the paper, we are more explicit regarding recommendations for future research (pgs. 9-11) as well as the limitations of the study (2nd paragraph, p. 11). Because we strictly adhered to NComms' formatting instructions, we did so in the Discussion section and did not add a separate concluding section.

R1.6: In general, the results of the manuscripts disaster frequency and loss are fundamental in shaping adaptation measures depending on the context of the city as determined by the existing complex relationships prevailing. Therefore, there is no universal common approach that is adopted by cities as an adaptative strategy but a piecemeal combination. The study results are fundamental in advancing research and debates in the climate change disaster and risk adaptation space especially in the global south which is not properly represented in the analysis.

Authors' response: This is an important point. Although the adaptation actions covered in our study were taken by cities from every major world region (see box b, Fig. 1, p. 2), over half were located in high-income countries, which is a constraint of our dataset. Nevertheless, our data does allow us to observe that income-level is an important confounder that drives both losses and adaptive capacity. The discussion section highlights the need for more research to investigate variations in disaster impacts on adaptation actions in low-income countries (see the highlighted text, p.).

NB; The highlighted aspects need to be addressed by authors to ensure that journal standards are maintained and met as required.

Authors' response: All of the highlighted aspects above have been addressed, as detailed in our responses.

Reviewer #2 (Remarks to the Author):

Recommendation: Major revision

R2.1. Thank you for the opportunity to review “Disaster frequency and loss shape hazard-related adaptations in cities.” This is a very interesting paper that addresses a major challenge and persistent gap for the adaptation literature – comparative explanatory research on adaptation responses, particularly at the local level. Nonetheless, there are two major issues here related to the time horizons in the dataset and the statistical models that have significant implications for the findings and require further attention. The following recommendations concern the formulation of the IV and DV data, and the types of models used in the analysis.

R2.2a. Time horizons: Why these particular years for the disaster data? The time horizon is very short – 1 year following the disaster period is likely too short in most cases to see the adoption and implementation of substantive adaptations, which typically take longer to achieve (e.g. such as actions 7-9 & 13, 14, 16, 18 in Table A1). Planning cycles alone generally run longer than 1 year for those types of initiatives- implementation can indeed take many years in the case of major adaptation projects like infrastructure construction or upgrades. This implies that many (perhaps most) adaptations occurring in 2018-2019 are not occurring in response to events that happened in 2015-2017, and/or that they were initiated prior to 2018. Furthermore, this creates a potential bias in the dependent variable, particularly for expansive adaptations that may take longer to plan and implement given the challenges of post-disaster recovery (which would be expected to focus on top-of-mind hazards rather than other potential hazards).

Authors' response: Thank you for these comments. We have taken several steps to address the suggestions regarding the time aspect as well as the issues concerning the statistical models. In response to the concern that the disaster period is too short, we have partially redesigned the analysis, which now models the impact of events occurring within a 5yr period prior to the year adaptation actions were reported within CDP. We do this for one CDP-year at the time, exploring impacts of disasters in 2013-2017 on CDP actions in 2018 and disasters in 2014-2018 on CDP actions in 2019, which we clarify on p. 3-4 (main text) and p. 12 (methods). While it is still plausible in theory that some adaptation actions originate in disasters further back in time, we hereby limit the risk of bias due to a short time horizon. These models also control for the number of years since the last disaster (p. 5, 14), and the results show no significant effects of time-lag on any of the adaptation action types.

R2.2b. Related to this, the application of the 15-year baseline here only measures more severe/frequent disaster events in 2015-2017. If there were more significant events in the

proceeding 5 years (that may have motivated adaptations implemented in 2018-2019), then it will seem as though cities actually have a lower frequency and severity of event experience in the current years of interests. The authors point to this as a limitation on page 12, however this is a significant issue (it speaks directly to the validity of the results) and it needs to be formally addressed in the analysis. What happens to the results if a longer time window of disaster data is used? E.g. 2010-2017 vs. a baseline of 2000-2010 (or even 2000-2017 and a baseline of 1990-1999).

Authors' response: This is addressed by the extended time-horizon described above, which means that we extended the period for calculating the baseline measures. Additionally, in preparing the revised version, we compared the results for the 15yr baseline measures with a 25yr baseline. As before, these measures are based on three-year running averages. The results show no differences between the 15yr and 25yr baseline measures – none of these have any significant effects on any of the adaptation action types.

R2.3. Why these particular years for adaptation data? Are the data from CDP only adaptations that were taken up in that period, or are they all the adaptations that were in place during that period but may have been adopted earlier (or that were just registered in the CDP database during those years)? If it's the latter, then those earlier adaptations should be removed if the DV focus remains on actions adopted in the 2018-2019 period.

R2.4. Overall the study needs more refinement around these issues, and sensitivity testing of the results to changes in the disaster events time horizon.

Authors' response: The years selected are due to the CDP data. We have responded to this concern, by adding a discussion (p. 11) acknowledging the absence of time-specific information in the CDP dataset, which does not provide any information to determine whether an action was adopted in the years prior to reporting or during the year of reporting, as an important data limitation. Consequently, we clarify in the methods section that adaptation actions include adopting new policy measures as well as the continuation of actions initiated in previous years (p. 12, under 'CDP'). We considered using information in the dataset concerning the "status of implementation" of any given action. However, this variable consisted of relatively rough pre-defined values (e.g., pre-feasibility study, scoping, pre-implementation, implementation, and operation) with no details about the timing of adoption and was also subject to a very large number of missing values.

R2.5. Models: There's a question of scale in the statistical models used here- the DV data is local, the disaster data is regional and national, and the adaptive capacity data is national. Why didn't the authors use multi-level modelling here to account for the hierarchical nature of the data (particularly the country-level indicators)? I advise them to run a separate set of MLMs and report on whether these findings differ from the current models.

Authors' response: This is an important point, and we prioritized addressing it in the revised version of the manuscript. Specifically, we dealt with the hierarchical nature of the data (local

DV data, regional and national disaster data, and national adaptive capacity data) by using cross-level interaction models (adding interaction-terms to multiple regression models) to explore potential mediating effects of adaptive capacity attributes. The results from this effort are presented in Table 2 (p. 7) and under the results section on p. 6-7. We then conducted a simple slopes analysis for each significant interaction term to unveil whether and how results differed across levels of adaptive capacity (results presented on p. 7 and in Fig 3, p. 8). This was an important addition to the analysis, which enabled us to identify several interaction-effects.

Additionally, we also addressed the question of scale by further clarifying in the revised version (p. 3) that our geocoding of the disaster data (EM-DAT) enabled us to link disasters to cities via regions (although, as we frankly acknowledge (p. 11), the exact location of events remains unspecified). Hence, we clarify throughout the paper (e.g. p. 4, 13, 14) that we focus on disaster impacts in ‘city regions’ or ‘regions where cities are located’ to recognize that in some cases cities may have been directly affected by a disaster but in other cases cities were indirectly affected by regional impacts.

R2.6. Second, why not control for local population, as it’s shown in the literature that large cities tend to do more adaptation (quantity of adaptation actions) and more substantive adaptation (e.g. Araos et al. 2017, Lesnikowski et al. 2020)? This pattern is found across high, middle, and low-income countries. Only controlling for GNI misses this variation within countries.

Authors’ response: This was a valuable suggestion, and in the revised version, we now include population as a control (p. 5). We implemented this suggestion by using the city population data provided by the CDP dataset. As we report in the revised version of the study (p. 6), the results show that city population indeed has a positive effect on several adaptation action types. Meanwhile, we did not find any interaction-effects between city population size and disaster frequency or severity, which we also report (p. 6).

R2.7. Other comments: Were adaptation actions related to non-climate change hazards removed from the dataset? On page 2 of the supplemental materials it’s implied that hazards/responses related to earthquakes and volcanic activity are left in. These should be removed from the dataset if this is the case (over-reporting is a common challenge with country-driven reporting on adaptation, so responses to those types of hazards may make it into the CDP database but aren’t aligned with conventional definitions of key climate risks, e.g. from the IPCC). If they were removed prior to analysis, then this should be specified on page 2.

Authors’ response: In the revised version (p. 13), we have clarified that the study includes all nine hazard event types used by the EM-DAT and specified in the supplementary material. As we explain, our rationale for keeping geological hazards in the analysis is that since adaptation actions in the CDP dataset are not exclusively limited to climate risks, we gain analytical leverage by being able to explore the possibility that experience from earthquakes or volcanos may provide incentives for measures to strengthen generic preparedness for

disaster (defined as ‘general adaptation actions’ in our study) or actions to address other social and environmental risks unrelated to climate change (‘other actions’). We realize that we did not explain this clearly enough in the first version of the paper. We have done so now, particularly in relation to the description of the adaptation action types in the introduction (p. 2).

R2.8. The paper observes that “increased economic losses from disasters generally are followed by an increase in the number of specific disaster adaptation actions. Increased economic losses are also associated with an increase in other climate adaptation actions unrelated to disaster preparedness (subcategory ‘other’). Economic losses do not influence the other adaptation action subcategories. These are robust effects that hold despite variation in adaptive capacity (Supplementary Table 21a)” (page 11). This is very interesting given previous research (e.g. Hallegatte et al. 2013) that finds economic damages are higher in developed countries and human damages are higher in developing countries, which is attributed to generally higher investment and resilience in infrastructure and social systems in wealthier places. Because the adaptive capacity data is national in this study, it’s difficult to draw conclusions from this difference in findings. Controlling for local population as suggested above would help to facilitate comparison to these other studies. Do economic losses drive specific disaster adaptation actions everywhere, or is this relationship being obscured here by the tendency for higher economic losses to occur in large cities with more costly infrastructure and economic activity?

Authors’ response: Thank you for highlighting this point, which is very relevant in light of the revised study. We provide (in the discussion section, p. 10) some descriptive information regarding the distribution of losses across income-levels in the study sample. In addition, following the other changes in the study design that we detailed above, we get slightly different results concerning the effects of economic damages on adaptation actions. Specifically, the moderation analysis (Table 2, p. 7) and the simple slopes analysis (Figure 3, p. 8) show that economic losses only affected certain adaptation actions (specific and general), and these effects only appear in cities located in countries with higher adaptive capacity. In the discussion section (p. 10), we suggest that this could be evidence of a wealth effect where developed communities both suffer greater monetary losses and have greater governance capacities for crafting adaptation actions.

R2.9. Supplemental materials addition: Please include a list or map of the cities in the dataset, and information on country/region membership.

Authors’ response: We have added a high-resolution map of the cities in the CDP dataset to the Supplementary Material file (see Supplementary Figure A1, p. 2). These derive from Fig 1 in the main paper and show the number of disasters and the number of adaptation actions by sub-national regions where the cities are located. In addition, we have added a complete list of all cities in a table in the Supplementary Material file (Supplementary Table A1).

Peer review comments second round -

Reviewer #1 (Remarks to the Author):

The comments were addressed very well but note the following:

- Insert in the technique/s used to explore the research.

-improve the resolution for figure 1.

In general, the manuscript has greatly improved in focus and clarity.

Reviewer #2 (Remarks to the Author):

Many thanks for the opportunity to review the updated version of this manuscript. I congratulate the authors on a thoughtful and thorough revision and am satisfied with the updates made to the analysis and discussion. I have no further comments and recommend the manuscript for publication. The analysis makes an important contribution to the scholarship and I look forward to seeing it in print.

Response to Reviewers:

Reviewer #1 (Remarks to the Author): The comments were addressed very well but note the following: - Insert in the technique/s used to explore the research. -improve the resolution for figure 1. In general, the manuscript has greatly improved in focus and clarity.

Authors' response: To address the first concern, the technique, i.e., the methods, used in the study is now clearly specified in the abstract. More detail on the techniques used also can be found in the methods section. To address the second concern, the resolution of figure 1 has been enhanced in accordance with the detailed instructions provided in the Author checklist. We are grateful for the comments by Reviewer #1, which were very useful in helping us improve the manuscript.

Reviewer #2 (Remarks to the Author): Many thanks for the opportunity to review the updated version of this manuscript. I congratulate the authors on a thoughtful and thorough revision and am satisfied with the updates made to the analysis and discussion. I have no further comments and recommend the manuscript for publication. The analysis makes an important contribution to the scholarship and I look forward to seeing it in print.

Authors' response: We are grateful for the comments by Reviewer #2, which were very useful in helping us improve the manuscript.